# Occurrence of *Aggregatibacter actinomycetemcomitans* and Its JP2 Genotype in a Cohort of 220 Western Australians with Unstable Periodontitis

**DOI:** 10.3390/microorganisms12112354

**Published:** 2024-11-18

**Authors:** Nabil Khzam, Omar Kujan, Dorte Haubek, Leticia Algarves Miranda

**Affiliations:** 1Dental School, The University of Western Australia, Nedlands, WA 6009, Australia; omar.kujan@uwa.edu.au; 2NK Periodontics, Specialist Periodontal Private Practice, Applecross, WA 6155, Australia; 3Jammerbugt Municipal Dental Service, Skolevej 1, DK-9460 Brovst, Denmark; dorte.haubek@outlook.dk

**Keywords:** *Aggregatibacter actinomycetemcomitans*, JP2 genotype, periodontitis

## Abstract

Aim: The main purpose of the present study was to investigate the carrier rate of *Aggregatibacter actinomycetemcomitans* and its JP2 genotype in a cohort of 200 Western Australians diagnosed with periodontitis. Materials and Methods: In this descriptive cross-sectional study, 220 consecutive patients with periodontitis, aged 18 years and older, were recruited to a specialist periodontal practice in Perth City. Every patient included in this study contributed three different intra-oral samples. Periodontal, radiographical, and microbiological assessments were performed. The samples were analysed using a polymerase chain reaction for the detection of *Aggregatibacter actinomycetemcomitans* and its JP2 genotype using the primers and conditions described previously. A Chi-square test and logistic regression analysis were performed to evaluate the results. Results: The prevalence of *Aggregatibacter actinomycetemcomitans* was 28.18%. The carrier rates of *A. actinomycetemcomitans* in the unstimulated saliva, cheek swabs, and pooled subgingival plaque samples were 21.80%, 19.50%, and 17.70%, respectively. There was a significant correlation between the severe form of periodontitis (stage IV, grade C) and younger age (*p* = 0.004), positive family history of periodontitis (*p* < 0.001), oral hygiene method (*p* < 0.001), and irregular dental visit attendance (*p* < 0.001). The binary logistic regression analysis revealed that having severe periodontitis risk increased almost three times in those who were young (OR: 2.812) and came from a family with a history of periodontal disease (OR: 3.194). However, the risk of severe periodontitis was five times higher in those patients with tooth loss due to periodontal disease (OR: 5.071). The highly leukotoxic JP2 genotype of *Aggregatibacter actinomycetemcomitans* was not detected. Conclusions: This study of a Western Australian cohort confirmed the low presence of *Aggregatibacter actinomycetemcomitans* and the complete absence of its JP2 genotype. Young age, family history of periodontal disease, lack of flossing, irregular dental visits, and tooth loss due to periodontitis were identified as potential risk factors for periodontitis stage IV, grade C in this cohort.

## 1. Introduction

Dysbiosis of the human microbiota can play an important role in a number of complex diseases, like diabetes mellitus, inflammatory bowel disease, and periodontitis [1]. The oral cavity is a home for more than 700 microbial species, forming one of the most complex and dynamic microbial communities in the human body [2]. Periodontitis is a common disease worldwide and is driven by a dysbiotic microbiota in genetically suspectable individuals [3]. Periodontitis is a chronic inflammatory disease that results in the loss of the tooth attachment apparatus and, if left untreated, can eventually lead to teeth loss.

A paradigm shift based on the current model of polymicrobial synergy and dysbiosis suggests that the progression of periodontitis is induced by a more comprehensive dysbiotic microbial community rather than by a selected number of periodontal pathogens [4]. The development of novel culture-independent techniques, such as PCR and next-generation sequencing, has identified several previously underappreciated bacteria, including the gram-positive *Filifactor alocis*, gram-negative *Aggregatibacter actinomycetemcomitans*, *Peptostreptococcus stomatis*, and other new emerging species.

*A. actinomycetemcomitans* is a small gram-negative, fastidious, capnophilic, facultatively anaerobic, non-motile, rod-shaped, non-sporing bacterium, which is related to the *Pasteurellaceae* family [5]. This bacterium was identified for the first time in 1912 in human actinomycosis lesions and was called *Bacterium actinomycetem comitans* [6]. In 1976, the World of Periodontology received the new addition of *Actinobacillus actinomycetemcomitans* as an associated microorganism in young people with periodontitis [7,8]. In 1979, the role of the bacterium in periodontitis started to become more clear as the extraction and partial characterization of the leukotoxin that was able to destroy human leukocytes was achieved [9]. This bacterium is genetically heterogeneous with different pathogenic potential. It has two genotypes, mainly JP2 and non-JP2, and seven serotypes (a to g) based on the O-polysaccharides [10,11,12]. Serotype b appears to be present in those patients with severe forms of periodontitis [13], especially in the virulent clonal lineage of *A. actinomycetemcomitans* serotype b, termed the JP2 genotype, which was characterised by a 530 basepair deletion in the promoter region of the leukotoxin gene apron. The JP2 genotype was originally isolated from young patients of African descent [14,15]. The JP2 genotype of *A. actinomycetemcomitans* is responsible for 10–20 times higher secretion of leukotoxins than the non-JP2 genotypes. The JP2 genotype of *A. actinomycetemcomitans* is associated with an increased prevalence of aggressive periodontitis in adolescents [15]. In the current study, we investigated the presence of *A. actinomycetemcomitans* and its JP2 genotype in 220 Western Australian periodontitis patients for the first time in the Oceania region using PCR methodology. The second aim was to assess potential indicators of the risk of severe form(s) of periodontitis in this cohort.

## 2. Materials and Methods

### 2.1. Study Population

In this descriptive cross-sectional study, 220 consecutive patients with periodontitis, aged 18 years and older, were recruited from two private periodontal practices in Perth City, Western Australia between June 2022 and June 2024 to take part in the present study. For more information regarding the recruitment process, the inclusion and exclusion criteria, sample size calculations, and the measurement reliability and reproducibility of the examiner, please refer to the previously published study [16]. Periodontal and radiographic examinations were carried out by one experienced periodontist (NK). In the present study, samples of unstimulated saliva (220 samples), cheek swabs (220 samples), and pooled subgingival plaque (1760) samples were collected from each subject. The pooled subgingival plaque samples were collected using a sterile universal mini-curette. For each patient, subgingival plaque samples were collected from four diseased and four healthy sites (two samples per quadrant). The total number of samples collected was 1760 subgingival plaque samples; these samples were pooled into 220 samples (1 sample per patient). About 2 mL of expectorated whole saliva was collected from each subject. From each patient, subgingival plaque samples were collected. Cheek swab samples were collected using a sterile plastic applicator. All samples were collected in sterile ice-chilled 5.0 mL Eppendorf tubes (Eppendorf South Pacific Pty, Ltd., Macquarie Park, Australia). Cases with periodontitis were defined according to the 2017 classification of periodontal and peri-implant diseases and conditions [17].

### 2.2. Bacterial DNA Extraction

For DNA isolation from the samples, a GXT NA Extraction Kit^®^ (Hain Lifescience, GmBH, Nehren, Germany) and an Arrow automated extraction instrument (Liaison IXT, DiaSorin Ltd., Dublin, Ireland) were used with procedures described earlier [18]. The process of the sample handling started with mixing 200 µL of the samples with 600 µL 1 M Tris buffer (pH 8.0) and DNA was extracted from 550 µL of this sample mixture and eluted in a volume of 100 µL. Suspensions of reference (HK1651) were treated and used for standard curves and serially diluted at different concentrations. The samples and the standard solutions were stored at 4 °C until use.

### 2.3. Bacterial DNA Quantification

The amount of total extracted DNA in each plaque sample was quantified using a NanoDrop (Thermo Fisher, Waltham, MA, USA) instrument. For the quantification of *A. actinomycetemcomitans* loads, suspensions of the reference strain HK1651 were treated and used for standard curves [18]. A Corbett Research Rotor-Gene 6000 Rotary Analyze instrument (QIAGEN, Valencia, CA, USA) was used for the quantification of the total concentration of *A. actinomycetemcomitans* loads in the samples, using qPCR. The cycling conditions used were according to the Kirakodu method (Table 1) [19]. The oligonucleotide primers used were as follows: forward 5′-CTAGGTATTGCGAAACAATTTG-3′ and reverse 5′-CCTGAAATTAAGCTGGTAATC-3′. A load of 100 *A. actinomycetemcomitans* cells per mL of sample was set as a positive result regarding the presence of this bacterium. The DNA concentration of *A. actinomycetemcomitans* in each sample was determined from duplicates.

### 2.4. Conventional PCR

For detection of the JP2 genotype, specific oligonucleotide primers targeting the leukotoxin promoter sequence of *A. actinomycetemcomians* were used [20]. Primers used for the detection of the ltx promoter gene of *A. actinomycetemcomitans* were forward: (GCCGACACCAAAGACAAAGTCT) and reverse (GCCCATAACCAAGCCACATAC) (Table 2).

### 2.5. Statistical Analysis

The collected data were analysed using IBM SPSS Statistics software program version 29.0 (SPSS Inc., Chicago, IL, USA). The primary outcomes were the presence of *A. actinomycetemcomitans* and the severity of periodontitis. A descriptive analysis was performed, evaluating quantitative and qualitative variables, and presented in tables. Furthermore, a two-tailed Chi-square test was carried out to evaluate the relationship between different levels of periodontitis, *A. actinomycetemcomitans* presence, and other potential risk factors, like age, gender, overall health, oral hygiene methods, frequency of dental visits, marital status, and smoking status. The confidence level was set at 95%. To assess the risk of an advanced form of periodontitis with the presence of the different potential risk factors evaluated, the methodology of binary logistic regression analysis was used. The significance level used was 5%.

### 2.6. Ethical Considerations

Those who agreed to participate in the present study signed an informed consent form. All patient data were entered into a cloud-based software program (Centaur Software Dental4Windows version) using coded numbers to protect their privacy. The approval to conduct the current study was given by the Human Ethics, Office of Research at The University of Western Australia (2022/ET000252).

## 3. Results

### 3.1. Presence of A. actinomycetemcomitans and Its JP2 Genotype

*A. actinomycetemcomitans* was present in 62 (28.18%) of the 220 participants and their levels of this bacterium in the analysed samples are presented in Table 3. The overall mean DNA concentration of *A. actinomycetemcomitans* in positive subjects from all sampled intra-oral sites was 6331.60 ng/μL (SD ± 4828.81) compared to 5.23 ng/μL (SD ± 1.48) in the negative subjects. The mean DNA concentration of *A. actinomycetemcomitans* in the unstimulated saliva samples was 4917.99 ng/μL (SD ± 3033.55) in *A. actinomycetemcomitans*-positive patients (*n* = 48). The mean DNA concentration of *A. actinomycetemcomitans* in cheek swabs was 851.47 ng/μL (SD ± 454.17) in *A. actinomycetemcomitans*-positive patients (*n* = 43). The mean DNA concentration of *A. actinomycetemcomitans* in pooled subgingival plaque samples was 13,225.33 ng/μL (SD ± 7743.19) in *A. actinomycetemcomitans*-positive patients (*n* = 39). The concentration of the *A. actinomycetemcomitans* in the older age group (subjects who are more than forty years old) was 22,332.65 ng/μL (SD ± 8650.9) versus 7345.21 ng/μL (SD ± 3075.58) in the younger age group (subjects who are forty years old and younger).

The numbers of positive subjects with *A. actinomycetemcomitans* in the unstimulated saliva, cheek swabs, and pooled subgingival plaque samples were 48 (21.80%), 43 (19.50%), and 39 (17.70%), respectively. More males were *A. actinomycetemcomitans*-positive compared to females (54.83% versus 45.16%). The youngest patient carrying *A. actinomycetemcomitans* was thirty years old and the oldest patient was seventy-six years old. The bacterium was present more often in the older age groups (53.20%). About 64.50% of non-Australian-born patients were *A. actinomycetemcomitans* positive. Most of the *A. actinomycetemcomitans*-positive subjects were diagnosed with generalised periodontitis stage III, grade B (67.74%). *A. actinomycetemcomitans*-positive subjects were 83.90% non-smokers, 56.50% living in Perth city, 87.10% and 69.40% low education levels and employed, 56.50% with family histories of periodontal disease, and 59.70% regular attendees to their dental appointments and used toothbrushing and dental floss to look after their oral health. Fifty of the *A. actinomycetemcomitans*-positive subjects had generalised periodontitis (80.60%) and had grade B periodontitis (74.20%) compared to sixteen with grade C periodontitis (25.80%). Stage III periodontitis was at 71.0% in the *A. actinomycetemcomitans*-positive subjects compared to stage IV at 24.20%.

### 3.2. Clinical Outcomes

A total of 220 patients were included in the present study; among these, 120 (54.20%) were females. The age ranged between 18 and 86 years old, with a mean of 48.63 ± 13.55 (SD). In total, 54.50% of the subjects were older than 40 years old, 18.60% were smokers and using electronic cigarettes, 68.20% were regular dental appointment attendees, 57.70% had a good oral hygiene routine with daily toothbrushing and flossing, and 56.40% had a family history of periodontitis. The reason for tooth loss in this cohort was due to periodontal disease in 40.50%. In the younger age group, 35.0% lost their teeth due to periodontitis. In the older age group, 45.0% lost their teeth due to periodontitis. Finally, among those patients with stage IV periodontitis, 69.60% lost their teeth due to periodontitis.

The overall mean PPD was 5.39 ± 2.19 mm while the overall mean CAL was 5.70 ± 2.11 mm. The distribution of the three PPD categories (<4 mm, 5 mm, and ≥6 mm) was as follows: 37.10%, 11.10%, and 51.70%. The overall BoP index and plaque index were 43.80% and 47.70%, respectively. Positive subjects with *A. actinomycetemcomitans* had deeper mean PPD values compared to those who did not harbour this bacterium (5.46 mm ± 2.12 versus 5.36 mm ± 2.13). The presence of *A. actinomycetemcomitans* was greater in those PPD ≥ 6 mm compared to those periodontal pockets that did not harbour this bacterium. The positive *A. actinomycetemcomitans* participants had 69.8% in the category of CAL ≥ 5 mm in the CAL group compared to 70.4% of those in *A. actinomycetemcomitans*-negative participants. The same closed figures are true for those CAL 1–2 mm and CAL 3–4 mm brackets between the positive and negative *A. actinomycetemcomitans* subjects (5.30%, 24.90% versus 6.60%, 22.90%) (Table 4).

### 3.3. Relationship Between Periodontitis and Potential Risk Factors

In total, 76.80% of the patients had generalised periodontitis. The distribution of grades A through C of periodontitis was grade A (0.90%), grade B (69.50%) and grade C (29.50%). The occurrence of the periodontitis stages was stage II (3.60%), stage III (70.90%), and stage IV (25.50%). Stage IV periodontitis did affect the younger patient group (34.0%) more than the older patient group (18.30%). Meanwhile, stage IV periodontitis almost equally affected the females and males (25.80% versus 25.0%). A total of 25.0% of the subjects were diagnosed with periodontitis stage IV, grade C, (females: 56.40% versus males: 43.60). In total, 60.0% of the younger age group were infected more than the older age group (40.0%). PPD ≥ 6 mm and CAL ≥ 5 mm are associated with stage IV grade C periodontitis, with these two categories affecting 58.20% and 61.80% of the subjects, respectively. Teeth that are lost because of periodontal disease are presented more in the severe periodontitis subjects (69.10%). A total of 87.30% of those who were diagnosed with severe periodontitis had a family history of the disease, 56.40% did not visit the dentist on a regular basis, and 63.60% did not floss (63.60%). Chi-square tests revealed a statistically significant association between periodontitis stage IV, grade C and age (*p* = 0.012), family history of periodontal disease (*p* < 0.001), dental visit attendance (*p* < 0.001), oral hygiene home care method (*p* < 0.001), and the reason of tooth loss (*p* < 0.001) (Table 5). The results of binary logistic regression, including variables with a potential relation to the presence of severe forms of periodontitis (stage IV, grade C), are presented in Table 6. An odd ratio was obtained for age groups (B = 1.034, OR = 2.812, CI: 1.151–6.867, *p* = 0.023), family history of periodontitis (B = 1.161, OR = 3.194, CI: 1.496–6.821, *p* = 0.003), and the reason of tooth loss (B = 1.624, OR = 5.071, CI: 2.247–11.447, *p* < 0.001).

## 4. Discussion

*A. actinomycetemcomitans* is associated with periodontitis and systemic diseases, such as infective endocarditis, bacteraemia, and meningitis [21]. In the present work, the overall *A. actinomycetemcomitans* prevalence was 28.18%. The prevalence of *A. actinomycetemcomitans* was high in those subjects living in Asia, both the Americas, and Africa but was significantly lower in the European population [22,23,24]. Our results are comparable to previous findings in some European populations. Our study is the first one in the Oceania region to determine the prevalence of *A. actinomycetemcomitans* in an adult population with unstable periodontitis. Our study analysed intra-oral samples taken from three different sources. All patients in our study were examined, diagnosed and classified according to the new 2017 periodontal and peri-implant diseases and conditions [17]. In our previous study, we included only 156 subjects, out of which 76.90% were older than 40 years old. As a result of this, we had to recruit another 64 patients of the younger age group to balance the age of the sample [16].

Few published studies have looked at the prevalence of *A. actinomycetemcomitans* in three intra-oral sites. In 2001, a study in Germany assessed the intra-oral distribution of *A. actinomycetemcomitans* in young adults with minimal periodontal disease [25]. Of the ninety-seven volunteers examined and assessed in this study, seventeen healthy volunteers (17.50%), being 20 to 27 years of age, harboured *A. actinomycetemcomitans* in pooled subgingival plaque, cheek mucosa and saliva samples. The samples were selectively cultivated for *A. actinomycetemcomitans.* In our study we found a higher prevalence than in this German study. This may be due to the higher detection ability of the PCR methodology. In another published study in Italy, the presence of *A. actinomycetemcomitans* using PCR in saliva and dental plaque samples that were collected was assessed. This study population consisted of eighty-one Italian (Sardinian) subjects with a mean age of 43.9 years old, among whom fifty-five (68%) had various clinical forms of periodontal disease [26]. A low number of Sardinian patients was found to be positive for *A. actinomycetemcomitans* in the oral cavity (12.34%). The carrier rate of the bacterium in this study was lower than in our study, despite including a significant number of patients with periodontitis. In a Swedish study, the prevalence of *A. actinomycetemcomitans* in periodontitis patients was assessed. The study included 3459 subgingival plaque samples taken from 1445 patients [27]. The population in this study was divided into 337 younger patients and 1108 older patients (the cut-off point was 35 years old). The bacterium could be isolated from around 30% of the sampled patients (bacterial culturing methodology was used). The prevalence of *A. actinomycetemcomitans* was higher among younger patients, which is opposite to the finding in our study. In the present study, the overall mean DNA concentration of *A. actinomycetemcomitans* in all intra-oral sampled sites was 6331.60 ng/μL (*n* = 62). The concentration of *A. actinomycetemcomitans* in the older age group was 22,332.65 ng/μL versus 7345.21 ng/μL. All these findings of our study are opposite to the Swedish study; perhaps taking subgingival plaque samples alone will not represent the real concentration and the presence of *A. actinomycetemcomitans*. The prevalence of *A. actinomycetemcomitans* in a Greek population was investigated in 2011 [28]. A total of 228 subjects participated in the study, each contributing one pooled subgingival sample analysed using PCR. The study included unstable and stable periodontitis; our study included only patients with unstable periodontitis. The outcome of this Greek study was that *A. actinomycetemcomitans* was detected more frequently in those patients who were diagnosed with unstable periodontitis (27.50%) compared with the other groups. The study included ninety-one subjects with a mean age of 51 years old, which was close to the age group included in our study. In our study, positive subjects with *A. actinomycetemcomitans* had deeper mean PPD values, CAL values, PPD ≥ 6 mm, and CAL ≥ 5 mm compared to those who did not harbour this bacterium but did not reach the statistical significance level.

None of the *A. actinomycetemcomitans*-positive subjects in the present study harboured the high-leukotoxin-producing JP2 genotype. This finding is in line with those observed in other Asian population studies assessing the presence of the JP2 genotype of *A. actinomycetemcomitans* [29,30,31,32]. The JP2 genotype of *A. actinomycetemcomitans* was not detected in the Danish study population, despite the fact that young subjects were included [23]. In another study in India with a much older population, the same negative outcome in regard to the JP2 genotype presence was found [24]. In Thailand, subgingival plaque samples from 453 subjects were analysed for the presence of the JP2 genotype but the genotype was not detected [33]. In a systematic review, the carriage rate of the JP2 genotype of *A. actinomycetemcomitans* was 6.37% in the world population with a high presence of the JP2 genotype found in subjects from South America, North America, and Africa [34]. Most of the studies assessing JP2 genotype presence analysed populations in a single country, particularly Morocco and Brazil [32,35]. There was a large number of studies that failed to detect the JP2 genotype of *A. actinomycetemcomitans,* which was a similar outcome to our study [36].

No edentulous patient was present in our study and the average number of teeth present was 30.8 teeth per patient. In the mainland China study, the mean number of teeth was 24.24 while in the United States, it was 24 teeth. In our study, there was less tooth loss [37,38]. In the HUNT4 study, the periodontal data outcome was as follows: PPD ≥ 4 mm was observed in 48.60% and PPD ≥ 6 mm was observed in 9.4%. Based on radiographic bone loss alone, 13.80% of participants defined as periodontitis cases were assigned to stage I, 44.80% to stage II, and 13.80% to stage III or IV. Grades A, B, and C were observed in 5.70%, 60.20%, and 6.20%, respectively [39]. In the SADLS studies, PPD ≥ 4 mm was observed in 62.88%, PPD ≥ 5 mm was in 27.24%, and, finally, PPD ≥ 6 mm was in 14.18% [40]. Periodontitis cases were assigned to stage I 4.23%, stage II 47.05%, and stage III or IV 46.10%. A recent Portuguese study revealed that the extent of periodontitis was classified as generalised in 66.5% of patients and as localised in 33.5% of patients [41]. In the Portuguese study, the prevalence of stage I was 6.10%; stage II was 12.30%; stage III, which was the most common, was 51.20%; and, finally, stage IV was 30.40%. In our study, 21.80% of the patients had localised periodontitis, 76.80% had generalised periodontitis, and a few had incisor–molar presentation (1.40%). We are the only group contributing with a study that has been published on the incisor–molar distribution of periodontitis according to the new classification of 2017. Our study grading was as follows: grade A periodontitis 0.90%, grade B 69.50%, and grade C 29.50%. In our current paper, there was no stage I disease in our sample. One reason for this could be that the patients that we had included were referred by their own dentist to seek our periodontal advice and treatment for more advanced forms of periodontitis. Stage II in our current paper affected 3.60% of the subjects, stage III was positive in 70.90%, and stage IV was present in 25.50% of the subjects. Our periodontitis staging figures were close to the Portuguese study [41]. Stage IV periodontitis did affect the younger patient group (34.0%) more often than the older patient group (18.30%). This is in agreement with the Portuguese study [41]. When it comes to gender, stage IV periodontitis almost equally affected the females and males (25.80% versus 25.0%). In our study, the mean PPD was 5.39 mm while the mean CAL was 5.70 mm. We could not compare these figures to other studies due to reasons like partial-mouth periodontal parameter recording in some of the studies while others mixed the healthy and diseased data together to make it extremely difficult to extract the exact periodontal measurements in those cases with active periodontitis [39,40,41]. In one of the Japanese cross-sectional studies that was published in 2020, an assessment of the periodontal conditions in the Takahagi population was conducted [42]. The study included 582 randomly sampled Takahagi residents aged between 20 and 89 years old. The mean PPD and CAL were 2.5 ± 0.5 mm and 2.9 ± 1.0 mm, respectively, which is much less than our periodontal parameters. The distribution of the three PPD categories in our study (<4 mm, 5 mm, and ≥6 mm) was as follows: 37.10%, 11.10%, and 51.70%, respectively. Our study had the highest PPD ≥ 6 mm compared to the HUNT4, SADLS, and Portuguese study. This can be explained by the fact that we included those patients who were referred to a specialist periodontal practice due to their initial diagnosis of severe periodontitis. The mean percentages of tooth surfaces harbouring plaque and exhibiting BoP were 59.50% and 31.10%, respectively [42]. Our study presented with a BoP value and plaque index of 43.80% and 47.70%, respectively. In a Portuguese population-based study, the BoP mean value was 26.16%, which is much less than our study [41].

A total of 25% of participants in this study had the most severe form of periodontitis, being stage IV, grade C. It affected more females (56.40%), those in the younger age group (60.0%), those who lost teeth due to periodontitis (69.10%), those with a positive family history of periodontal disease (87.30%), those with a lack of dental flossing (63.60%), and irregular attendees to dental apportionments (56.40%). It was not possible to compare the above-mentioned data with other studies due to the lack of published materials, combined with the scarce data of epidemiological figures on periodontal diseases based on the 2017 classification scheme. One of the few studies that we could use in this discussion was the mainland China study [37]. More than 30% of the Chinese adults suffered from severe periodontitis (stages III and IV). Data from the Study of Health in Pomerania showed that the prevalence of severe periodontitis was 17.60% [43]. In a sample of the US population (2011–2012 National Health and Nutrition Examination Survey), the overall prevalence of severe periodontitis was 13.2% [44]. In the HUNT4 study, the prevalence of stage IV periodontitis was 2.3% but there was no grading combined with the severity [39]. In 2019, a study was carried out to explore the prevalence of periodontitis in adults who live in the Southern region of Lisbon in Portugal [45]. The prevalence of severe periodontitis (stage III and IV) in this Portuguese population was 24.0%, which is closer to our study.

Our data showed a higher risk of severe periodontitis (stage IV, grade C) in the younger age group. This is in contrast with other studies that showed an increase in the severity of periodontitis in the older age group [37,41,43]. The advanced form of the disease is also known to be more common in men than women, which is opposite to our findings, where females had more advanced periodontitis compared to their male counterparts [37,41,43]. However, some of those studies used a half-mouth recording periodontal protocol, which can contribute to disease underestimation and inaccuracy; also, they did not include enough young patients with periodontitis in their sample and, finally, the difference in the used classification and case definition of periodontitis was another issue.

In our study, there was a statistically significant association between the most severe form of periodontitis (stage IV, grade C) and age groups, family histories of periodontitis, dental visit attendance, oral hygiene methods, and the reasons for tooth loss. Those who were diagnosed with an advanced form of periodontitis were in the younger age group and had a positive family history of periodontal disease, irregular dental care, lack of dental flossing, and lost their teeth mainly due to periodontal disease. The results of binary logistic regression in our research, including variables with a potential relation to the presence of severe forms of periodontitis (stage IV, grade C), resulted in three statistically significant relationships with age group, family history of periodontal disease, and the reason for tooth loss. The binary logistic analysis revealed that the younger age group patients had almost three times the risk of having a severe form of periodontitis compared to those in the older age group (OR = 2.812, CI: 1.151–6.867, *p* = 0.023). Those patients with a family history of periodontitis had more than three times the risk of having a severe form of periodontitis compared to those with no family history of periodontitis. (OR = 3.194 CI: 1.496–6.821, *p* = 0.003). Those patients who lost their teeth due to periodontal reasons were five times more likely to have an advanced form of periodontitis. (OR = 5.071, CI: 2.247–11.447, *p* < 0.001). In a study by Relvas et al. in 2022, a high prevalence of periodontitis, using a binary logistic regression, was reported on. They noticed a significant relation between the risk of periodontitis and increased age (OR = 1.033, CI: 1.005–1.062, *p* = 0.019), which is opposite to our findings with a severe form of periodontitis in the younger age subjects [41]. Relvas et al. also found that the lack of tooth brushing (OR = 0.25, CI: 0.105–0.599, *p* = 0.002) and lack of dental flossing (OR = 0.63, CI: 0.09–0.768, *p* = 0.015) were a possible risk factor for periodontitis. It was concluded that the increased age and lack of tooth brushing and flossing were identified as potential risk factors for periodontitis in the investigated Portuguese population. In another logistic regression analysis to ascertain risk factors towards periodontitis, the risk of periodontitis significantly increased with age (OR = 1.05, CI: 1.04–1.06), active and former smokers (OR = 3.76 and OR = 2.11), those with lower education levels (OR = 2.08, OR = 1.86, for middle and elementary education, respectively), and those with diabetes mellitus (OR = 1.53) [45].

Limitations of our study are the lack of control groups (healthy periodontium), the inability to detect a cause–effect relationship, and the representative results being unable to be fully guaranteed due to the snapshot nature of the sampling methodology. All patients were referred to periodontal practice, thus, most of them were periodontally unstable and needed periodontal treatment (non-surgical, surgical, and laser therapy) [46]. The convenience sampling method utilised indicates that the periodontitis experienced by the participants was more likely to be higher than in a randomly selected study population.

## 5. Conclusions

This study revealed a relatively low presence of *A. actinomycetemcomitans* in a cohort of Western Australia with a complete absence of the JP2 genotype. Potential risk factors for an advanced form of periodontitis in this cohort were identified as young age, family history of periodontitis, and tooth loss due to periodontal disease. More research will be needed to explore the important and potential role that *A. actinomycetemcomitans* can play in the diagnosis, treatment, and potential vaccination of periodontitis in the Australian population.

## Figures and Tables

**Table 1 microorganisms-12-02354-t001:** DNA quantification of *A. actinomycetemcomitans*.

	Forward	Reverse
Kirakodu (*ltxA*)	CTAGGTATTGCGAAACAATTTG	CCTGAAATTAAGCTGGTAATC

Primers used for qPCR-based quantification of *A. actinomyetemcomitans* in accordance with the Kirakodu method.

**Table 2 microorganisms-12-02354-t002:** Conventional PCR of the JP2 genotype of *A. actinomycetemcomians*.

	Forward	Reverse
Poulsen *ltx-promoter*	GCCGACACCAAAGACAAAGTCT	GCCCATAACCAAGCCACATAC

Primers used for the detection of the *ltx* promoter gene of *A. actinomycetemcomitans* in accordance with the Poulsen method.

**Table 3 microorganisms-12-02354-t003:** *A. actinomycetemcomitans*-positive patients and their characteristics.

Patient/Variables	Sex	Age in Years	Origin	Stage/Grade	*Aa* Saliva ng/μL	*Aa* Cheek Swabs ng/μL	*Aa* Plaque ng/μL
Pt 1	F	70	Philippines	III/B	72,055	-	848,041
Pt 2	F	32	Philippines	III/C	-	119	-
Pt 3	M	30	Australia	III/B	-	1956.5	-
Pt 4	M	40	Australia	IV/B	-	511	-
Pt 5	F	63	Australia	III/B	-	219	-
Pt 6	M	48	Australia	IV/C	19,950.5	2679	22,379
Pt 7	F	56	England	III/B	1075	161	-
Pt 8	F	48	Australia	III/B	606.50	-	10,401
Pt 9	F	40	New Zealand (Māori)	IV/C	1470	638.5	1719
Pt 10	M	49	New Zealand (Māori)	III/B	8530	-	1680
Pt 11	M	56	Thailand	III/B	14,290	165.5	1448
Pt 12	M	54	New Zealand (Māori)	III/B	2500	-	303
Pt 13	M	35	Australia	III/B	3810	-	-
Pt 14	M	68	Australia	IV/C	9820	570.5	365,681
Pt 15	F	51	Australia	II/B	5124	-	-
Pt 16	M	34	Tonga	IV/C	8685	2299.5	851
Pt 17	M	57	Canada	III/B	6865	-	-
Pt 18	F	56	England	III/B	-	-	153
Pt 19	F	65	New Zealand (Māori)	IV/C	13,910	247.5	10,651
Pt 20	M	57	Australia (Aboriginal)	III/B	4115	-	-
Pt 21	M	39	England	III/B	2415	-	577
Pt 22	F	40	Australia	III/B	26,620	1219.5	153
Pt 23	M	57	Australia	III/B	2065	-	-
Pt 24	M	33	Australia	III/C	1710	-	-
Pt 25	M	64	Australia	III/B	1050	-	-
P 26	F	71	Ireland	IV/C	1325	-	-
Pt 27	F	62	Kenya	III/B	76,820	2716	381,842
Pt 28	F	36	Australia	III/B	2195	559	4624
Pt 29	M	40	Australia	III/B	-	-	340
Pt 30	M	57	Samoa	III/B	1255	204	56,334
Pt 31	F	47	Palestine	III/B	411,735	604.5	46,227
Pt 32	M	40	Australia	III/B	54,790	153	184,822
Pt 33	M	75	Italy	III/B	-	-	132
Pt 34	M	56	South Africa	III/B	1510	-	479,910
Pt 35	M	52	Sri Lanka	III/B	3360	22,932	5163
Pt 36	F	40	Philippines	IV/C	-	105	-
Pt 37	F	52	England	III/B	120,725	23,264.5	-
Pt 38	F	60	South Africa	IV/C	33,910	25,995	254,437
Pt 39	M	76	Ireland	III/B	-	105	-
Pt 40	F	69	England	III/B	-	133.50	-
Pt 41	M	62	England	III/B	-	945.5	-
Pt 42	F	40	Ireland	III/B	4965	43,559	81,934
Pt 43	F	32	Australia	III/B	54,440	139.50	-
Pt 44	M	49	England	III/B	13,495	-	-
Pt 45	F	49	Greece	IV/C	46,010	503.5	-
Pt 46	M	68	Australia	III/B	-	353	3598
Pt 47	M	37	England	IV/C	-	106.5	-
Pt 48	M	33	Australia	III/B	4095	30,455.5	101,611
Pt 49	M	63	Australia	III/B	-	114.5	-
Pt 50	M	66	Australia	III/B	1525	-	138.50
Pt 51	M	40	China	IV/C	564	237	262
Pt 52	F	38	Australia	II/B	8845	3782	3978
Pt 53	M	38	New Zealand (Māori)	III/B	12,377	5135	5432
Pt 54	M	33	Vietnam	IV/C	5567	4822	19,792
Pt 55	F	36	Libya	III/B	4133	1103	1281
Pt 56	M	40	Australia	III/B	1298	856	1234
Pt 57	F	39	Poland	III/B	1593	1104	3789
Pt 58	F	39	Australia	III/B	2267	1409	1817
Pt 59	F	32	New Zealand (Māori)	IV/C	1621	1381	627
Pt 60	F	36	Italy	IV/C	656	263	449
Pt 61	M	38	New Zealand	III/B	3250	3063	5343
Pt 62	F	32	Italy	III/B	996	429	419

Pt: Patient, M: Male, F: Female.

**Table 4 microorganisms-12-02354-t004:** Summary of the distribution of categorical variables for periodontal and radiographic parameters related to gender, age, and presence of *A. actinomycetemcomitans*.

Variables		BoP (%)	PI (%)	Pus (%)	BL/Age > 1 (%)	BL > 33 (%)	V-BL (%)	PA Lesion	Endo-Perio Lesion
Sex	M	48.70	53.50	3.30	45.50	46.60	32.30	28.80	22.10
	F	39.80	43.0	4.0	53.0	54.80	47.80	25.90	22.0
Age	≤40 Yrs	39.20	43.90	2.90	47.10	46.20	47.80	23.50	20.50
	>40 Yrs	48.10	51.20	4.40	52.0	55.80	34.50	30.60	23.40
*Aa*	Present	47.40	52.10	6.30	46.60	46.70	35.10	30.80	19.90
	Absence	42.40	46.0	2.60	50.90	52.90	43.20	25.70	22.90

%: Percentage, ±SD: standard deviation, M: male, F: female, BoP: Bleeding upon Probing, PI: Plaque Index, Yrs: Years, BL: bone loss, V-BL: vertical bone loss, PA: periapical, Endo-Perio: Endodontic-Periodontal.

**Table 5 microorganisms-12-02354-t005:** Shows stage IV, grade C periodontitis and other variables.

Variables		Periodontitis Staging and Grading		X^2^	*p*-Value
		Stage IV, Grade C N (%)	Other Stages, Grades N (%)		
Age group	Younger age group	33 (60.0)	67 (40.60)	6.258	0.012
	Older age group	22 (40.0)	98 (59.40)		
Family history of periodontitis	Yes	48 (87.30)	76 (46.10)	30.233	<0.001
	No	4 (7.30)	78 (47.30)		
Dental visits attendance	Regular	24 (43.60)	126 (76.40)	20.366	<0.001
	Irregular	31 (56.40)	39 (23.60)		
Oral hygiene method	Toothbrushing and flossing	20 (36.40)	107 (64.80)	13.715	<0.001
	Toothbrushing only	35 (63.60)	58 (35.20)		
Reason of tooth loss	Periodontal	38 (69.10)	51 (30.90)	25.147	<0.001
	Non-periodontal	17 (30.90)	112 (67.90)		

N: Number, %: Percentage.

**Table 6 microorganisms-12-02354-t006:** Results of binary logistic regression analysis of variables with a potential relation to the presence of severe forms of periodontitis (stage IV, grade C).

Variables							95% CI for OR	
	B	S.E.	Wald	df	*p*	OR	LL	UL
Age groups	1.034	0.456	5.148	1	0.023	2.812	1.151	6.867
Family history of periodontitis	1.161	0.387	9.001	1	0.003	3.194	1.496	6.821
Reason of tooth loss	1.624	0.415	15.275	1	<0.001	5.071	2.247	11.447
Constant	−5.576	3.124	3.185	1	0.074	0.004		

B: estimates for the slope coefficients of the univariate logistic regression model containing only this variable, S.E.: estimated standard error for the estimated coefficient, df: degree of freedom, *p*: value associated with the statistical coefficient test, OR: estimated odds ratio, CI: confidence interval of 95% for odds ratio, LL: lower limit, UL: upper limit.

## Data Availability

The raw data supporting the conclusions of this study will be made available by the authors.

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
