# Peer review of "Occurrence of Aggregatibacter actinomycetemcomitans and Its JP2 Genotype in a Cohort of 220 Western Australians with Unstable Periodontitis"

_microorganisms, 2024, doi:10.3390/microorganisms12112354_

Round 1

Reviewer 1 Report

Comments and Suggestions for Authors

The work of Nabil Khzam et al, regarding the prevalence of Aggregatibacter actinomycetemcomitans and its JP2 genotype in western Australians raises some interesting findings. Nevertheless, I found the manuscript lacking coherent structure between the different sections, as well as a poor presentation of the data, which significantly impacts the clarity of the message and the significance of the findings.

I ask authors to include line numbering in their peer review file, as this makes it easier to point out reviewers' suggestions in a concise manner.

2. Materials and methods

• Why do the authors refer to reproducibility and essential details for a separate manuscript, compared to another already published one? In this case, why did the authors not publish all the findings in a single manuscript? This raises serious doubts about the legitimacy and ethics of the manuscript. In addition, they continuously cite reference 16, which is a previous work by the same authors.

• The authors talk about “pooled gingival plaque”, but there are no details of this, how was it taken? In addition, they talk about 1760 samples, please explain if each patient provided a number n of samples.

• Bacterial quantification: The authors must include the details of experimental reproducibility in this manuscript, even if they have used the technique previously. Above all, because it is a key technique for the entire work.

• Statistical analysis: The authors do not point out normality tests for their data, nor do they detail whether the analyses were taken with one or two tails.

3. Results

• The way the results are written is excessively repetitive, convoluted and not very understandable.

• The units in which they present the DNA concentration are strange. Do they refer to DNA/uL of what exactly? How did they make the standard curve? Why didn't they take CFU/uL? Or why didn't they use synthetic genes for a standard curve?

• The authors mention that the DNA concentration of A. actinomycetemcomitans in positive subjects is 6,331.60 ng/μL ± 48,288.01. Why do they present such a high SD? It is much higher than the reported mean. This also occurs in non-stimulated saliva.

• When the authors talk about contrasts in numerical values, they mention that there are differences, but they never mention the p value. It is important to point this out in order to know whether or not the comparison they make is statistically significant. (Eg. PPD measurements compared to females (5.41 mm ± 2.09 versus 5.37 mm ± 2.27)”

• Why are tables 4 and 5 relevant? How do they contribute to the manuscript?

• The authors make continuous statements, such as “The most common extent and presentation of periodontitis in this cohort was the generalized pattern affecting 76.80% of the patients.” What is this “generalized pattern”?

• The authors show tables in a continuous manner, in which individual values ​​are shown. This is not correct, since the aim is to perform statistical analyses of the population, so they should present graphs or tables that condense the mean values ​​(or medians, depending on the type of data), and then generate the hypothesis contrast. Ask for support from a statistics expert.

• It strikes me that the authors do indicate the p significance values ​​in section 3.3, where they obtain values ​​< 0.05, why in the comparisons 4. Discussion

• The section that begins with “None of the A. actinomycetemcomitans positive subjects in the present study…” is too long, and does not contribute anything to the manuscript, especially considering that no patient in the cohort was positive for the JP2 genotype.

• The authors mention that “In our study patient’s gender was used to compare the periodontal and radiographic clinical data. We found that the males had a slightly deeper PPD, worse CAL, higher BoP index, higher plaque index, more endodontic-periodontal lesions compared to females”, however, there are no statistically significant differences that support these conclusions. I do not think it is correct to indicate a result of this type as a relevant finding.

In summary, although the topic addressed by the authors is interesting, it seems to me that the manuscript is poor in terms of form, structure and presentation of data, which negatively impacts the relevance of the conclusions. Furthermore, the way in which they present some data makes one suspect that they are trying to highlight non-significant results as key points of the work, when this is not the case, which in turn calls into question the ethical aspects of the presentation of data.

Author Response

  1. Materials and methods
  • Why do the authors refer to reproducibility and essential details for a separate manuscript, compared to another already published one? In this case, why did the authors not publish all the findings in a single manuscript? This raises serious doubts about the legitimacy and ethics of the manuscript. In addition, they continuously cite reference 16, which is a previous work by the same authors.

I am sorry for the confusion this resulted in. In our first study we recruited 156 patients, however I could not find any JP2 genotype and I though because I did not include enough young patients in the samples (40 years old and younger). As result I had to start all over again and recruit another new 64 patients who were 40 years old and younger, to ensure that my sample was balanced age-wise. I have no intentions whatsoever to cast any doubts about the legitimacy and the ethics of the current work. This was presented in lines 178-180.

  • The authors talk about “pooled gingival plaque”, but there are no details of this, how was it taken? In addition, they talk about 1760 samples, please explain if each patient provided a number n of samples.

Line 63-65, the pooled subgingival plaque samples were collected using a sterile universal mini-curette. For each patient, subgingival plaque samples were collected from four diseased and four healthy sites (two samples per quadrant). The total number of samples collected was 1760 subgingival plaque samples; these samples were pooled into 220 samples (one sample per patient).

  • Bacterial quantification: The authors must include the details of experimental reproducibility in this manuscript, even if they have used the technique previously. Above all, because it is a key technique for the entire work.

This is added now in lines 72-76 and lines 82-85.

  • Statistical analysis: The authors do not point out normality tests for their data, nor do they detail whether the analyses were taken with one or two tails.

This is added now, line 93.

  1. Results
  • The way the results are written is excessively repetitive, convoluted and not very understandable.

All the result section is amended (red coloured text).

  • The units in which they present the DNA concentration are strange. Do they refer to DNA/uL of what exactly? How did they make the standard curve? Why didn't they take CFU/uL? Or why didn't they use synthetic genes for a standard curve?

The DNA concentration unit used in the study is ng/μL.

  • The authors mention that the DNA concentration of A. actinomycetemcomitansins positive subjects is 6,331.60 ng/μL ± 48,288.01. Why do they present such a high SD? It is much higher than the reported mean. This also occurs in non-stimulated saliva.

This is amended now (lines 106-108).

  • When the authors talk about contrasts in numerical values, they mention that there are differences, but they never mention the p value. It is important to point this out in order to know

whether or not the comparison they make is statistically significant. (Eg. PPD measurements compared to females (5.41 mm ± 2.09 versus 5.37 mm ± 2.27)”

This was removed from the result section.

  • Why are tables 4 and 5 relevant? How do they contribute to the manuscript?

Table 4 and 5 both were removed.

  • The authors make continuous statements, such as “The most common extent and presentation of periodontitis in this cohort was the generalized pattern affecting 76.80% of the patients.” What is this “generalized pattern”?

There are three patterns in periodontitis according to the 1999 and the 2017 classifications (localized, generalized, incisor/molar), this is changed now (lines 121-122, 156).

  • The authors show tables in a continuous manner, in which individual values are shown. This is not correct, since the aim is to perform statistical analyses of the population, so they should present graphs or tables that condense the mean values (or medians, depending on the type of data), and then generate the hypothesis contrast. Ask for support from a statistics expert.

Those tables were removed.

  • It strikes me that the authors do indicate the p significance values in section 3.3, where they obtain values < 0.05, why in the comparisons

This is corrected now (line 167).

  1. Discussion
  • The section that begins with “None of the A. actinomycetemcomitans positive subjects in the

present study…” is too long, and does not contribute anything to the manuscript, especially

considering that no patient in the cohort was positive for the JP2 genotype.

This section was shorten (lines 189-198).

  • The authors mention that “In our study patient’s gender was used to compare the periodontal and radiographic clinical data. We found that the males had a slightly deeper PPD, worse CAL, higher BoP index, higher plaque index, more endodontic-periodontal lesions compared to females”, however, there are no statistically significant differences that support these conclusions. I do not think it is correct to indicate a result of this type as a relevant finding.

This was removed

Reviewer 2 Report

Comments and Suggestions for Authors

please see the enclosed pdf

Author Response

1- Introduction: the author should mention the main methods of bacteria identification.

This is added to the introduction section lines 51-52.

2- Discussion: the author should mention methods to inactivate Aa in periodontal treatment such as laser.

This is added to the discussion section lines 266-268

3- References: reference are old and very limited in number, this section needs improvement.

Reference list was screened and adjusted accordingly.

Reviewer 3 Report

Comments and Suggestions for Authors

Can be accepted with minor corrections and text editing

Author Response

1- Abstract: PPD, CAL, PPD: write full form along with abbreviation.

These were removed now.

2- Elaborate introduction please.

This is amended now.

3- Aa is gram negative and should not be listed with gram positive.

This is amended now, line 37.

4- Replace with JP2.

This is amended now, line 48.

5- Remove this.

This is amended now, line 48.

6- Change to – from.

Amended now, line 55.

7- Remove t.

Amended now, line 77.

8- Add to.

Amended now, line 113.

9- Move percentage.

Amended now lines 114-119

10- Change to limitations.

Amended now line 265.

Reviewer 4 Report

Comments and Suggestions for Authors

Decision: Reconsider after major revisions

The paper investigates the prevalence of Aggregatibacter actinomycetemcomitans (Aa) and its highly leukotoxic JP2 genotype among a cohort of 220 Western Australians diagnosed with unstable periodontitis. Utilizing a descriptive cross-sectional study design, the authors employed polymerase chain reaction (PCR) to detect the presence of Aa and its genotypes from various intraoral samples. Key findings include a 28.18% prevalence of Aa, absence of JP2 genotype, and significant associations of Aa presence with severe periodontal disease markers like greater pocket depth and clinical attachment loss. The paper underscores potential risk factors, including young age, familial history, and poor oral hygiene practices, thereby contributing to the understanding of periodontitis' microbial and epidemiological aspects in the Australian context.

Specific comments:

Abstract:

1) Please clarify what is meant by ‘consecutive’. Was this a truly consecutive sample, or were there specific inclusion/exclusion criteria that might have introduced selection bias?

2) The abstract reports several statistically significant associations in this section. However, it does not present any actual data (e.g., odds ratios and confidence intervals). Please include these data points to strengthen the abstract.

3) The abstract abruptly concludes that the JP2 genotype was not detected. Consider briefly mentioning the implications of this finding and its potential significance.

Introduction

1) The introduction lacks a clear research question or hypothesis. Clearly state the primary aim of the study and the specific research questions being addressed.

2) This paragraph describes the JP2 genotype and its association with aggressive periodontitis. However, it needs more context within the overall study. How does investigating the JP2 genotype in this cohort contribute to the understanding of periodontitis in Western Australians?

Method

1) No control group of periodontally healthy individuals was included, which may limit the understanding of Aa’s prevalence exclusively within periodontitis patients versus a normal population.

2) The study mentions a focus on ‘unstable periodontitis’ but does not clearly define the criteria for instability, which might lead to variability in patient selection and results interpretation.

Results

1) The phrase "mostly nonsmokers" is vague. Provide the exact percentage of smokers and nonsmokers within the Aa positive group for accuracy.

2) This section describes the periodontal and radiographic findings in relation to Aa presence. However, the findings are presented as a series of observations without clear interpretation. Discuss the significance of these findings, particularly in the context of the existing literature.

3) The abstract reports a prevalence of 28.18% for Aa. However, it also provides separate percentages for saliva, cheek swabs, and plaque samples. Clarify whether the 28.18% refers to the overall prevalence or a specific sample type.

Discussion

1) This section discusses the statistically significant association between periodontitis and various risk factors. However, the discussion is largely descriptive. Provide a more critical analysis of these associations, exploring potential causal links and considering alternative explanations.

2) The discussion of periodontal parameters in relation to gender and age lacks depth and clarity. Refine this section to provide a more focused and insightful analysis of these relationships.

3) The discussion of the JP2 genotype being absent from the study cohort is limited. Expand this section to further explore the potential reasons for this finding and its implications for understanding aggressive periodontitis in the Western Australian context.

Conclusion:

1) The conclusion reflects the study’s aims but could expand on the potential clinical implications or future research directions indicated by the findings.

Overall, the manuscript needs substantial revisions to enhance clarity, conciseness, and the depth of discussion. By addressing the specific comments outlined above, the authors can significantly strengthen the manuscript and enhance its contribution to the field of periodontology.

Author Response

Abstract:

1) Please clarify what is meant by ‘consecutive’. Was this a truly consecutive sample, or were there specific inclusion/exclusion criteria that might have introduced selection bias?

Yes, we saw the patients in consecutive manner, as long as they fit within our criteria to be included in the present study.

2) The abstract reports several statistically significant associations in this section. However, it does not present any actual data (e.g., odds ratios and confidence intervals). Please include these data points to strengthen the abstract.

This is amended now, lines 18-22

3) The abstract abruptly concludes that the JP2 genotype was not detected. Consider briefly mentioning the implications of this finding and its potential significance.

 Due to word count restriction in the abstract section, I could not do this, I am sorry.

Introduction

1) The introduction lacks a clear research question or hypothesis. Clearly state the primary aim of the study and the specific research questions being addressed.

This is updated now, lines 52-53.

2) This paragraph describes the JP2 genotype and its association with aggressive periodontitis. However, it needs more context within the overall study. How does investigating the JP2 genotype in this cohort contribute to the understanding of periodontitis in Western Australians?

 We included in our study patients diagnosed with stage-III/IV periodontitis, ruling out the presence of JP2 clone will help us focus on other microbial/genetic/environmental factors that can contributes to the severe forms of periodontitis, which is covered partially by the second aim of the present study.

Method

1) No control group of periodontally healthy individuals was included, which may limit the understanding of Aa’s prevalence exclusively within periodontitis patients versus a normal population.

Yes, this is one of the limitations of this study, this was addressed in the limitations part of the text, line 267. We are a periodontal practice, and we see patients with periodontitis most of the time.

2) The study mentions a focus on ‘unstable periodontitis’ but does not clearly define the criteria for instability, which might lead to variability in patient selection and results interpretation.

 We used the case definition of periodontitis according to the 2017 classification, which cover the term unstable periodontitis (active disease), lines 67-68.

Results

1) The phrase "mostly nonsmokers" is vague. Provide the exact percentage of smokers and nonsmokers within the Aa positive group for accuracy.

83.90% were non-smokers and 16.10% were smokers, line 116-117.

2) This section describes the periodontal and radiographic findings in relation to Aa presence. However, the findings are presented as a series of observations without clear interpretation. Discuss the significance of these findings, particularly in the context of the existing literature.

This was covered in many details at the discussion section, lines 164-188.

3) The abstract reports a prevalence of 28.18% for Aa. However, it also provides separate percentages for saliva, cheek swabs, and plaque samples. Clarify whether the 28.18% refers to the overall prevalence or a specific sample type.

Yes, the 28.18% is referring to the overall prevalence of Aa

Discussion

1) This section discusses the statistically significant association between periodontitis and various risk factors. However, the discussion is largely descriptive. Provide a more critical analysis of these associations, exploring potential causal links and considering alternative explanations.

This is amended now, lines 246-269.

2) The discussion of periodontal parameters in relation to gender and age lacks depth and clarity. Refine this section to provide a more focused and insightful analysis of these relationships.

This is removed from the text due to lack of any statistical significance.

3) The discussion of the JP2 genotype being absent from the study cohort is limited. Expand this section to further explore the potential reasons for this finding and its implications for understanding aggressive periodontitis in the Western Australian context.

 This is amended now, lines 189-198.

Conclusion:

1) The conclusion reflects the study’s aims but could expand on the potential clinical implications or future research directions indicated by the findings.

This amended now.

Round 2

Reviewer 1 Report

Comments and Suggestions for Authors

Dear authors, I see a significant improvement in the clarity of the article, purpose and presentation of the data. It seems to me that it now has the potential to be published.

However, I still see certain features that need to be corrected:

Uniform the terms: concentrations, loads per ng/uL or abundance terms when discussing bacterial counts.

Line 168: German study maybe? is it correct to spell "Germany study"?

Table 1 and 2: I do not see the need to insert a table for the PCR conditions. Please review the style of other molecular microbiology reports.

Table 3: It would be much more useful and concise for the reader to make a table in which the mean or median of the variables is expressed, instead of displaying a large table in which the variables of the captured population cannot be observed punctually.

Table 4: Were statistical tests of hypothesis contrast between the variables performed? For example, BoP with Aa present vs BoP with Aa absent? If applicable, include the significance values ​​obtained in the table.

Author Response

Uniform the terms: concentrations, loads per ng/uL or abundance terms when discussing bacterial counts.

This is amended now, lines 105-111

Line 168: German study maybe? is it correct to spell "Germany study"?

Changed, line 168

Table 1 and 2: I do not see the need to insert a table for the PCR conditions. Please review the style of other molecular microbiology reports.

Both tables modified as requested, lines 357-371

Table 3: It would be much more useful and concise for the reader to make a table in which the mean or median of the variables is expressed, instead of displaying a large table in which the variables of the captured population cannot be observed punctually.

I feel that the table is very informative and showing the exact and accurate information regards the subjects, I hope you don’t mind me keeping it. Thanks.

Table 4: Were statistical tests of hypothesis contrast between the variables performed? For example, BoP with Aa present vs BoP with Aa absent? If applicable, include the significance values â€‹â€‹obtained in the table.

Non of the values reached statistical significance in this table.

Reviewer 2 Report

Comments and Suggestions for Authors

The authors did not update the referance list, the discussions section is too short and the authors did not compare their results with the literature, the discussion regarding laser should have a referance, I suggest : Martu, M.-A.; Luchian, I.; Mares, M.; Solomon, S.; Ciurcanu, O.; Danila, V.; Rezus, E.; Foia, L. The Effectiveness of Laser Applications and Photodynamic Therapy on Relevant Periodontal Pathogens (Aggregatibacter actinomycetemcomitans) Associated with Immunomodulating Anti-rheumatic Drugs. Bioengineering 2023, 10, 61. https://doi.org/10.3390/bioengineering10010061

Author Response

The laser reference was added, lines 277-280. There were few studies with published data using the 2017 periodontal disease and conditions classifications, hence the shorter discussion.

Reviewer 4 Report

Comments and Suggestions for Authors

The manuscript has significantly improved and reads much more coherently. The introduction is clearly written and provides a strong rationale for the study. The methods section is more detailed and well-organized.  The revised results and discussion sections nicely tie together the main findings. Consider briefly mentioning any future research directions in the conclusion.

Author Response

This is added now, lines: 284-286